# Monthly and Spatially Resolved Black Carbon Emission Inventory of India: Uncertainty Analysis

**Umed Paliwal**[1]**, Mukesh Sharma**[1]**, and John F. Burkhart**[2,3]

[1]Department of Civil Engineering, Indian Institute of Technology Kanpur 208016, India
[2]Department of Geosciences, University of Oslo, Norway
[3]Sierra Nevada Research Institute, University of California, Merced, California, USA

*Correspondence to:* J. F. Burkhart
(john.burkhart@geo.uio.no)

**Abstract.** Black Carbon (BC) emissions from India for the year 2011 is estimated to be 901.11± 151.56 Gg/yr based on a new ground-up, GIS-based inventory. The grid-based, spatially resolved emission inventory includes, in addition to conventional sources, emissions from kerosene lamps, forest fires, diesel-powered irrigation pumps and electricity generators at mobile towers. The emissions have been estimated at district level and were spatially distributed into grids at a resolution of 40x40 km2. The uncertainty in emissions has been estimated using Monte Carlo simulation by considering the variability in activity data and emission factors. Monthly variation of BC emissions has also been estimated to account for the seasonal variability. To the total BC emissions, Domestic Fuels contributed most significantly (47%), followed by Industry (22%), Transport (17%), Open Burning (12%) and Others (2%). The spatial and seasonal resolution of the inventory will be useful for modeling BC transport in the atmosphere for air quality, global warming, and other process-level studies that require greater temporal resolution than traditional inventories.

## 1 Introduction

Carbonaceous aerosols, defined as Black Carbon (BC) also known as Elemental Carbon (EC) and Organic carbon (OC) (Pachauri et al., 2013), form a significant and highly variable component of atmospheric aerosols. Neither BC nor OC has a precise chemical definition. OC includes numerous organic compounds, some of which are found to be carcinogenic, such as Poly Aromatic Hydrocarbons (PAHs) (Menzie et al., 1992; Pedersen et al., 2005). The Intergovernmental Panel on Climate Change (IPCC) defines BC as "Operationally defined aerosol species based on measurement of light absorption and chemical reactivity and/or thermal stability" (IPCC, 2013). BC is released from incomplete combustion of carbonaceous fuels such as agricultural and forest biomass, coal, diesel, etc. The type of combustion greatly affects the BC emission rates, notably; inefficient combustion emits more BC than efficient combustion for the same type of fuel. Aside from air quality and health effects, there are a number of climate impacts of BC emissions including alterations to temperature through atmospheric adsorption, modifications to precipitation timing, increased melting of snow (Meehl et al., 2008; Flanner et al., 2007; Ramanathan and Carmichael, 2008; Quinn et al., 2007; Koch and Del Genio, 2010; Bond et al., 2013) all of which are consequential to global warming. BC has been implicated to be the second largest contributor to global warming after CO2 (Ramanathan and Carmichael, 2008). There is a current debate that due to the short life span of BC, the BC atmospheric concentration will drop quickly if emissions are reduced, thereby potentially offering a rapid means to slow down global warming (Bond and Sun, 2005; Grieshop et al., 2009; Kopp and Mauzerall, 2010; Bowerman et al., 2013).

India is a rapidly growing economy with massive future growth potential. The total energy and coal consumption has almost doubled from 2001 to 2011 (IEA, 2012). The emissions of particulate matter or aerosols have been rising over the last few decades and are expected to increase in the future as well, due to rapid industrial growth and slower emission control measures (Menon et al., 2010). Recent studies (Yasunari et al., 2013; Lau et al.) have shown that the deposition of BC in the Himalayan glaciers has accelerated their melt-

ing. While BC is a source of warming at global scale, at a regional scale, it has adverse effects on air quality and human health. BC is a major part of particulate matter with size less than 2.5 micron (PM2.5), and like other PM2.5 particles, it is small enough to be inhaled. According to the World Health Organization (WHO), exposure to BC can lead to cardiopulmonary morbidity and mortality. WHO also suggests that BC may act as a universal carrier of chemicals of varying toxicity to lungs (Janssen et al., 2012). Understanding the sources of BC, their emissions and spatial distribution is important both for policy making and improving climate modeling. Preparation of an accurate emission inventory is the first step towards developing robust air pollution control strategies. Air quality measurement stations are installed at limited locations and are unable to provide a measure of spatial variability. However, observations coupled with air quality models, can provide comprehensive information about the impact of various sources on ambient air quality and their spatial variability. The greatest benefit of these models is gained after preparing an accurate emission inventory, validating the models with observations, thereby enabling a tool for improved control measures.

Although there have been several emission inventories developed for BC in the last decade, the estimates are variable without any knowledge of uncertainties. Model-predicted BC concentrations over India are 2 to 6 times lower than the observed concentrations (Ganguly et al., 2009; Nair et al., 2012; Bond et al., 2013; Moorthy et al., 2013). Further the current estimates vary considerably; Reanalysis of tropospheric chemical composition (RETRO) emission inventory (Schultz et al., 2007, 2008) estimated BC emissions in 2010 as 697 Gg/yr, the System of Air Quality Weather Forecasting and Research (SAFAR) emission inventory (Sahu et al., 2008) estimated it as 1119 Gg/yr for year 2011, Klimont et al. (2009) reports BC emissions as 1104 Gg/yr for year 2010 and Lu et al. (2011) reported it as 1015 Gg/yr for year 2010. Not only there is a need to get a meaningful total estimate but also to assess the uncertainty and spatial variability associated with these estimates. Most of the emission inventories provide yearly emissions and do not account for sub-annual temporal emission variability which leads to inaccurate impact assessments. To improve the nature of advanced numerical forecasts of impacts from aerosol pollution, we have developed an emission inventory at monthly resolution.

The objective of this study is to prepare a sub-annual, high spatial resolution, comprehensive spatially gridded emission inventory of BC emissions for India for the base year 2011. The approach is a ground-up inventory based on activity data from various sectors, combined with emission factors. While results are provided for one year, the frequency and distribution should be general enough such that coupled with growth forecasts, multi-year use could be valid. In this study, we have prepared a district-wise emission inventory available on a $40 \times 40 km^2$ grid. We have accounted for all the major sources of BC emissions in India. For example, emissions

from kerosene lamps (Lam et al., 2012) and forest fires which were previously unaccounted for in many emission inventories have been included. Monthly variation of BC emissions has also been estimated to provide better input for air quality models. We employ a unique approach to quantify uncertainty in the emissions by considering variability in (i) activity data from various sources, and (ii) emission factors (EFs). Specifically, probabilistic distributions were assigned to both activity data and EFs. By employing Monte Carlo simulation, several activity level and EFs were generated to arrive at emissions (by multiplying generated activity data and EF) which could be interpreted in terms of a mean value and associated uncertainty.

In Section 2 we present the methods used in our analysis, Section 3 describes the source sectors and activity data we considered, a description of the magnitude of emissions from each sector is presented in Section 4.

## 2   Methods

Our approach may be divided into two parts. Figure 1 presents the methodology for developing national emissions and their uncertainty and Figure 2 presents the approach for extracting gridded emissions. For estimating national emissions, a thorough review of multiple national activity data and EFs for each source was conducted from available published and unpublished sources (Table 1 and Table 2).

We fit a probability distribution function (PDF) to both national activity data and EFs from a pool of distributions on the basis of Kolomogorov-Smirnov test (KS statistic) using Mathwave Technologies EasyFit© software (Mathwave Technologies, 2015). Using the optimal probability distribution function (PDF) for both variables (EFs and activity data) for each source, we generated 1000 estimates of each variable from each of the two distributions. Further increasing the number of generations did not change the mean and the variance of the emissions.

For activity data that had only one source of information, a normal distribution with mean as the data point and standard deviation of 20% of the data point was assumed based on the experience of other data sets (Table 1). Best fit distributions were only determined from KS statistic if the number of data points exceeded 5, in other cases, a uniform distribution was assumed.

For preparing the gridded inventory, the emissions were first estimated within a Geographic Information System (GIS) using polygons at the district level. Polygons were subsequently dividing into $40 \, x \, 40 \, km^2$ grid elements, and were proportionally assigned emissions based on area. The area for grid elements spanning a district border was accounted for. Emissions from Industry (point data) were added directly to the overlying grid based on available location coordinates for the source. For the Road Transport (network) sector, the data from at the district level was distributed along the road

network and then assigned to overlying grids proportionally to the length of road in the grid element. Interpolation of the data was not conducted, as this would lead to erroneous georeferencing of emissions, particularly in the case of point data. More details are found in the subsections below.

For the national level annual inventory, Monte Carlo simulations were undertaken to specifically estimate mean emissions and uncertainties, whereas, at the district level the mean of the EFs and district level activity data were used to arrive at average emission levels. An image of the political map of India (Census of India, 2011) was georeferenced using Google Earth and 640 districts were digitized as polygons to generate a national level shape file. This shape file had an attribute table containing all the districts, and yearly emission quantities were recorded for each district. The shapefile and polygon data were resampled to a $40 \times 40 km^2$ grid by calculating the area of each portion of the districts within a grid element, and attributing that portion of the emissions to the grid. As a grid cell may overlay over more than one district, the overall emission in each cell was calculated by summing up part of emissions from each contributing portion from the district, based on area of the district within the grid cell and emission density for the district:

$$E_{cell} = \sum_{i=1}^{n}(\rho_i \cdot A_i) \tag{1}$$

Where $n$ is the total number of districts within each grid cell, $\rho$ is the Emission Density ($g/sm^2$) for each district and $A$ is the Area of the district ($m^2$) within the grid. Emission density (mass/time-area) was calculated by dividing the BC emission in the district with the total area of the district.

## 3 Source Sectors and Activity Data

The emissions sources considered in this study can be broadly categorized into five sectors: Open Burning, Industry, Transport, Domestic Fuel, and Others. In the following section we define the activity data and emission sources considered within each sector. All the emission sources identified by Reddy and Venkataraman (2002a, b) and Sonkar (2011) were included in this study. Also some of the highly emitting sources identified in the recent literature (Kerosene lamps, Diesel generators, and Irrigation pumps) were also considered. Table 1 and 2 provide an overview of activity data and EFs for the sources considered.

### 3.1 Open Burning

The open burning sector includes: forest fire emissions, open solid waste burning, and agriculture residue burning.

#### 3.1.1 Forest Fire

According to the 2013 Forest Survey of India (FSI), around 50% the forest area of India is prone to forest fires (FSI, 2013). There is a strong seasonality associated with forest fires in India with majority of fires occurring in the months from February to July. The causes of forest fire in India are both man-made and natural; natural causes being the high temperature and low humidity. Man-made causes include accidental fires and forest burnt for shifting cultivation. The forest fire burnt area in this study was determined using MODIS (Moderate Resolution Imaging Spectroradiometer) monthly burnt area product MCD45A1 which has a resolution of 500m (Land Processes Distributed Active Archive Center (LP DAAC), 2000). MODIS product MCD12Q1 (500m resolution) was used to define forest cover. The burnt area and land cover products were retrieved from LP DAAC website (https://lpdaac.usgs.gov/).

The methodology used for emission estimation is presented in Figure 3. Burnt area (MCD45A1) and land cover product (MCD12Q1) are available in Hierarchical Data Format - Earth Observing System (HDF-EOS) format and have an Earth gridded tile area of 1200 km x 1200 km. They were stitched to cover the whole geographical extent of India. The stitched products were converted to GeoTIFF image format and clipped to Indian domain using the ESRI shape file of the boundary of India. The same methodology was used for the burnt area product as well as the vegetation cover. Monthly burned area GeoTIFF images were overlayed on land cover image to determine the monthly forest burnt area pixels, and subsequently the forest area burnt. Dry mass per unit area of forest burnt was taken to be 5.2 kg/m2 (Joshi, 1991). Emissions were distributed district-wise according to the number of incidents of forest fire occurring in that district in 2011. The data of district-wise incidents of forest fire were taken from the most recent forest survey (FSI, 2015). Figure 4 shows the land cover image and burnt area image used for estimating the forest fire burnt area in January 2011. It can be noted that the emissions from this subsector can easily be updated for the future years using the latest MODIS burnt area and land cover products and following the aforementioned methodology.

#### 3.1.2 Municipal Solid Waste Open Burning

The dry weight content of Indian municipal solid waste (MSW) was estimated using the MSW composition in India (CPCB, 2007) and the dry matter content of MSW components per IPCC 2006 (2006). Indian MSW is primarily composed of vegetables (40%), stones (42%) and grass (4%) which has a dry matter content of 40%, 100% and 40% respectively. Dry matter content was estimated to be 67.6%.

State-wise MSW generated and collected was derived from the Central Pollution Control Board (CPCB) Municipal Solid Waste Management Report 2012 (CPCB, 2012).

The MSW generated was distributed among districts according to their urban population. For the states where MSW collected volume was not available, a value of 60% of the total MSW generated was assumed (Kumar, 2010). The total MSW openly burned was taken to be 10% of the collected waste and 2% of the uncollected waste (National Environmental Engineering Research Institute (NEERI), 2010). To provide a second approach for the uncertainty analysis, per capita waste generation in India and fraction burnt was taken from IPCC 2006 (2006). The 2011 census population data were used to provide the urban population of the district. From this, the total MSW burned for each district was taken as the product of the IPCC guideline results and the urban population.

### 3.1.3 Agricultural Residue burning

India generates a large amount of agricultural residues (e.g. waste biomass) every year after harvesting crops. These residues are used as domestic and industrial fuel, fodder for animals, etc. but a large amount remains unutilized in the fields. The quickest and easiest way for the farmers to manage this waste is to burn it. Figure 5 shows a flowchart for estimating emissions from crop residue burning.

The state-wise production of cotton, Jowar, Barley, Jute, Ragi, Rice, Maize, Bajra, Groundnut, Sugarcane, Wheat and Rapseed and Mustard in 2011 was taken from the Ministry of Agriculture, 2012 (www.apy.dacnet.nic.in). The crop production was distributed among districts of that state according to the net sown area (Ministry of Agriculture, 2011) in that district. Emission from crop residue burning was calculated using the following equation as suggested by Jain (2014).

$$ECRB = \sum_{i=1}^{D} \sum_{j=1}^{C} (P \cdot Q \cdot R \cdot S \cdot T \cdot EF_{BC}) \qquad (2)$$

where, ECRB is the Emissions from crop residue burning. The summation is done over the Districts, $D$ and for each type of Crop, $C$. The emission is then calculated from the product of Crop production, ($P$), residue to crop ratio, ($Q$), dry matter fraction ($R$), the fraction burnt ($S$), the fraction actually oxidized, ($T$), and finally the EF for BC. Three estimates of crop residue burnt ($P \cdot Q \cdot R \cdot S \cdot T$) were obtained by varying $Q$, $R$, and $S$, while holding $P$, $T$ constant for all the three estimates. In the first estimate, residue to crop ratio($Q$), dry matter fraction ($R$) and fraction burnt ($S$) was taken from Jain (2014). In the second estimate, residue to crop ratio and dry matter fraction was kept same and fraction burnt was taken as 0.25 for all the crops (IPCC 2006, 2006). In the third estimate, residue to crop ratio and dry matter fraction was taken from Venkataraman et al. (2006), and fraction burnt was kept as 0.25 (IPCC 2006, 2006). This provided us with three estimates of the total crop residue burnt in the fields (Table 1).

## 3.2 Industry

The Industrial sector includes: brick production, cement, steel plants, sugar mills and power plants. In general emissions and activity data for these sectors are derived from available published reports and scientific literature. We then use location information from each of the facilities to develop district-wise emissions. In order to construct the gridded inventory, industrial units were geolocated precisely using the provided GPS coordinates wherever available. In general, geolocated coordinate data is available for Iron & steel manufacturing, cement, sugar mills, and power production. Where exact information regarding facility locations cannot be obtained directly, the district-wise distribution is a function of population density. Within the Industry sector, this is the case for Brick Kilns, adding a source of uncertainty to the analysis but also a novel emission, for which prior studies have not included.

### 3.2.1 Brick Industry

The Indian Brick industry has more than 100,000 brick kilns producing 250 billion bricks and consuming about 25 million tons of coal annually (Gupta and Narayan, 2010; Maithel et al., 2012). Bricks in India are produced locally in small enterprises at rural scale (Rajarathnam et al., 2014). It is a seasonal industry operating predominantly from months of October to June (Maithel et al., 2012). Brick kilns can be classified into two major categories based upon firing practice: intermittent and continuous kilns. Intermittent kilns include clamp, Scove, Scoth and Down Draft Kilns (DDK) in these kilns bricks are fired in batches. In Continuous kilns brick heating and cooling takes place simultaneously in different parts of the kiln. Several types of kilns including: Bull's Trench Kiln (BTK), Hoffman Kiln, ZigZag Kiln, Tunnel kiln and vertical shaft brick kiln (Heierli and Maithel, 2015) operate continuously.

In India a majority of the bricks are produced from Fixed Chimney Bull's Trench Kiln (FCBTK) and clamps (Rajarathnam et al., 2014). There are around 60,000 small scale clamp kilns in India. Located all over India – mostly near or in villages and using biomass, coal, and lignite as fuel (Rajarathnam et al., 2014) these represent an important source of BC emissions. No account of their location, production, fuel consumption and Emission factors have been published. For this study, emissions only from FCBTK's are used which account for 70% of the total bricks produced from India and use coal as the primary fuel (Weyant et al., 2014). The state-wise brick production (in Kg) through these kilns was compiled from consultation with industry experts. It was distributed district-wise according to the population of the districts in the state. The quantity produced was assumed to be normally distributed with 50% standard deviation (Maithel et al., 2012).

### 3.2.2 Cement Manufacturing

The plant-wise cement production in 2011 was taken from Cement Manufacturers Association, Govt. of India (CMA). India had around 150 major cement plants in 2011, which produced 180 million tons of cement and consumed 28 million tons of coal. Cement being a transport-expensive product, plants are evenly distributed across India. Since the plant-wise coal consumption was not available, the national consumption by cement industry was taken from the same source. The fuel consumption was distributed using available location data and based on cement production in 2011.

### 3.2.3 Iron & Steel Production

India produced 68.6 million tons of total finished steel in 2010-11 consuming 40 million tons of coal (Ministry of Steel, 2014). The plant-wise steel production was taken from Press Information Bureau (2011), Govt. of India. The coal consumption was distributed among plants according to their level of steel production. District-wise coal consumption in steel plants was subsequently determined from the location of these plants.

### 3.2.4 Sugar Mills

India ranks second globally in terms of sugar production. Significant BC emissions result from sugar mills due to the usage of bagasse as a fuel. Bagasse is the fibrous residue obtained from sugarcane juice extraction and consists of cellulose (50%), hemicellulose(25%) and lignin(25%) (Ezhumalai and Thangavelu, 2010; Abhilash and Singh, 2008). India has a total of around 550 sugar mills which produced 26.3 million tons of sugar by crushing 361 million tons of sugarcane (Indian Sugar Mills Association (ISMA, 2012; DAC, 2013). Specific geolocated data is available and was used to distribute the emissions in the gridded data set. The mill-wise sugarcane crushing capacity was taken from the Department of Food & Public Distribution (DFPD, 2011). The total sugarcane crushed was distributed among mills according to their crushing capacity. The bagasse generated was taken as 30% of the total sugarcane crushed (Pessoa Júnior et al., 1997).

### 3.2.5 Power Plants

The Indian Central Electricity Authority (CEA, 2012) reports the plant-wise fuel consumption for coal and diesel power plants in India. In 2011, India had a installed capacity of 112 GW of coal and 1.2 GW of diesel based thermal power plants. There are around 100 coal based and 14 diesel based major thermal power plants located across India with location data available from government reports. District-wise fuel consumption was estimated by the location of these plants using the data contained in the report.

### 3.3 Transport

From the transportation sector emissions from road vehicles, railways, shipping, and aviation have been accounted for individually. For road vehicles, to prepare gridded data from district level emissions, road network data from OpenstreetMap ©. (OpenStreetMap, 2016) was utilized. The data provides a high resolution road network in vector format. The district shapefile, grid polygons and road network shapefile were resampled to a $40 \times 40 km^2$ grid by calculating the total road length in each portion of the districts within a grid element, and attributing that portion of the emissions to the grid. For non-road vehicles, methodology as discussed in section 2 was used.

### 3.3.1 Road Vehicles

Road vehicles have been divided into seven categories: two wheelers, cars, light motor vehicles (LMV), light commercial vehicles (LCV), taxies, trucks, buses, tractors and trailers.

State-wise number of registered vehicles in the aforementioned categories were taken from Ministry of Road Transport and Highways (2011). The vehicles were distributed among districts of that state according to the population of that district. In determining the emissions for 2011, we needed an estimate of the number of vehicles on road for that year. The reported number of registered vehicles represents the cumulative number of first registrations (Parikh and Radhakrishna, 2005). In India, vehicles are neither deregistered when they are no longer in use nor are double registrations deducted. The actual number of vehicles on the road is significantly smaller than the number of registered vehicles. Baidya and Borken-Kleefeld (2009), determined the rolling fleet in 2005 using survival functions. The category-wise number of vehicles on road as a fraction of registered vehicles was taken from Baidya and Borken-Kleefeld (2009). Emissions from road were estimated using the number of vehicles on road and the annual distance traveled per vehicle type.

$$EV_{district} = \sum_{i=1}^{n}(N_i \cdot AKT_i \cdot EF_i) \qquad (3)$$

Where $EV$ is the total BC emissions from vehicles for a district ($g/district/year$), $i$ is type of vehicle, $N$ is the number of vehicles, $AKT$ is the annual kilometer travelled for the vehicle type ($km/year$), and $EF$ is the vehicle type emission factor ($g/km$).

Annual average distance traveled is difficult to quantify and is a source of uncertainty in the emissions. The annual distance traveled by various vehicle types was derived from seven different studies (Table 1). This provided us with multiple estimates of total distance travelled by a vehicles type in a year. For some vehicle types only few BC EFs were available, to compensate for lack of information, EFs were de-

rived from PM2.5 emission factors using BC/PM2.5 fraction given by Chow et al. (2011).

### 3.3.2 Railways

Railways in India are primarily powered by electricity and Diesel, the use of coal has decreased over the years and is negligible now. Annual report (2010-11) of Indian railways enumerates the consumption of diesel and coal (Ministry of Railways, 2012b). The state-wise allocation of fuel consumed was performed according to the railway track length in the state (Ministry of Railways, 2012a) and finally district wise according to the population of the district.

### 3.3.3 Shipping

The Ministry of Petroleum and Natural Gas (MoPNG) reports the total consumption of Fuel Oil (FO), High-speed Diesel Oil (HSDO), and Light Diesel Oil (LDO) by the shipping subsector in 2011 (MoPNG, 2014). According to IPCC guidelines (IPCC 2006, 2006) the fuel used in international bunkers is not counted in the national emission inventory and their estimate is recorded separately. The proportion of shipping fuel used domestically was assumed from the European Environment Agency (EEA, 2013). Due to the non-availability of a spatial proxy, the emissions from ships have not been distributed district-wise and have only been accounted for in the national emissions.

### 3.3.4 Aviation

The total Aviation Turbine Fuel (ATF) consumption in India was taken from MoPNG (2014). Domestic operations accounts for 38% of the total fuel consumption (ICAO, 2010). Domestic fuel consumption was divided into that used for Landing and Take-off (LTO) and for cruise operations. The Directorate General of Civil Aviation (DGCA) reports the total number of domestic scheduled and non-scheduled aircraft departures in 2011 (DGCA, 2013). The fuel consumption per LTO was taken from IPCC 2006 (2006). The LTO ATF consumption was distributed district-wise according to the number of flights landing and taking off from the airports in that district. The cruise emission was not distributed and was only counted in national emissions.

### 3.4 Domestic Fuel

The Domestic Fuel sector includes: emissions from firewood, agricultural residue, coal, liquid petroleum gas (LPG), kerosene (cooking and lighting) and dung cake.

India faces a crucial challenge of providing clean and affordable energy sources to its rural households, especially in the cooking sector. Eighty-five percent of the rural households are still dependent upon traditional biomass fuel for their cooking needs (MoSPI, 2014b). Figure 6 shows the distribution of rural households on the basis of energy source used for cooking (MoSPI, 2014b).

The stoves used for cooking are inefficient causing incomplete combustion hence releasing more BC than would result from efficient combustion. In the year 2000, domestic fuels contributed 64% to the total BC emissions in Asia (Streets et al., 2003). State-wise per capita consumption (rural and urban) of firewood, LPG, coal was taken from National Sample Survey (MoSPI, 2014b) which releases a report of household consumption of various commodities using a large sample of households every 5 years. Apart from this, Yevich (2003) report the state-wise total consumption of firewood, agriculture residue and dung cake in 1985. We extrapolated the fuel consumption data to 2011 by using the growth rate of rural population from 1985-1991 and change in number of households using these fuels for cooking from 1991 to 2011. Smith et al. (2000) also report the state-wise consumption of firewood, dung cake and agricultural residue in 1991. We extrapolated the data to 2011 using the change in number of households using these fuels for cooking from 1991 to 2011. Using data from MoSPI (2014b), Yevich (2003), and Smith et al. (2000), three estimates of domestic fuels consumed in 2011 were prepared and used within the uncertainty analysis.

According to the World Bank (2010), 25% of the Indian population does not have access to electricity. As a result Kerosene fueled lamps are the only source of lighting after sunset for a large part of the population. In 2011, over a billion liters of kerosene was consumed to fuel these lamps (MoSPI, 2014b). The information on kerosene consumed was available from two sources: MoSPI (2014b) and Lam et al. (2012). The National Sample Survey (MoSPI, 2014b) reports the state-wise per capita (rural and urban) kerosene consumption. The proportion of kerosene used for cooking versus lighting in India was taken from Lam et al. (2012). Another estimate of kerosene consumed in lamps was derived following the methodology described in Lam et al. (2012).

### 3.5 Other

Others sector incorporates emissions from use of diesel in power generation sets. One of the largest consumers of diels are Irrigation Pumps. In addition, diesel is used in power generation for mobile towers, private households, small industry and commercial enterprises.

### 3.5.1 Irrigation Pumps

Agriculture is a core economic activity of India with about 60% of the population involved in the activity. In 2011 India used around 2.4 billion liters of diesel for irrigation pumps (MoPNG, 2013). The use of dug wells and tube wells is very common for irrigation purposes in India. Diesel powered pumps are used for mini irrigation schemes in farms with minimal or no access to electricity. The diesel consumed

was distributed district-wise according to the net sown area in that district (Ministry of Agriculture, 2011).

### 3.5.2   Diesel Generator Sets

In 2011-12, India faced an overall power deficit of 8.5% and peak power shortage of 10.6% (CEA-LGBR, 2013). To deal with this deficit there were prolonged power cuts throughout the country especially during the peak consumption period. Increasingly private households, commercial enterprises and industries are using diesel generators to maintain consistent power supply during power outages. Although there is no official estimate of the amount of diesel consumed by diesel generators, ICF International estimates that 4.51 billion liters of diesel was used in year 2012-13 (Shakti Sustainable Energy Foundation, 2014) . The growth rate of power deficit in India was used to adjust this value for 2011 (CEA-LGBR, 2013). Telecom industry is one the largest users of diesel generators. In 2011, India had more than half a million cell towers (Press Information Bureau, 2011). Most of these towers are located in villages where grid-connected electricity is not available. They use small generators fueled by diesel for their power needs. The total diesel consumption by cell towers was taken from MoPNG (2013). The fuel consumed was distributed state-wise according to the number of mobile towers in that state. It was then distributed district-wise according to the population of the district. Diesel consumed by generators in mobile towers was deducted from total amount of 4.51 billion liters consumed by diesel generators to estimate the remaining amount. Due to the paucity of data it was not possible to spatially distribute the remaining emissions to grids, so they have only been accounted for in the total national emissions.

## 4   Results and Discussion

Tables 1 and 2 present the probabilistic best fit distributions, mean and standard deviation for activity data and EFs for sources considered in this study. The mean district level activity data and EFs were used to estimate the district-wise emissions. It may be noted that kerosene lamps have the highest EF among all sources considered in this study; these lamps convert 8.5% of the fuel directly into BC. In the Open Burning sector, forest fires have the highest EF. In the Industry sector, EF is highest for sugar industry, as the industry uses bagasse as a fuel in a very inefficient combustion process. For Transport sector, EF for diesel operated vehicles (railways, ships, bus, truck, tractor & trailer, LCV) are higher than that for gasoline operated vehicles (two wheeler, LMV, car).

Total BC emissions for year 2011 have been estimated to be 901 ± 152 Gg (Table 3) of which 47% (425 Gg) originated from Domestic Fuel consumption, 22% (198 Gg) from Industry, 17% (154 Gg) from the Transport sector and 12%

(103 Gg) from Open burning. Diesel use in mobile towers and irrigation pumps contributed 2% (20 Gg) to total BC emissions (Figure 7). Firewood with emissions of 177 Gg is the single most emitting source. It emits more than transportation (154 Gg) and open burning (103 Gg) categories. As shown in Figure 6, 76.3% of the 140 million rural households MoSPI (2014b) use firewood as the primary source of energy for cooking.

The spatial distribution of national emissions is presented in Figure 8. From the map it can be easily concluded the IGP is the main contributor to national BC emissions. This can be attributed to the very high population density and presence of major BC emitting industries like sugar and brick production in this region. Some of the states in IGP are among the least developed in India with little access to even basic amenities like electricity, clean cooking fuels, sanitation, healthcare, etc. More than 90% of the rural households in Uttar Pradesh and Bihar use biomass fuels as their primary source of cooking and more than 65% are dependent upon kerosene lamps as their primary source of lighting (NSSO, 2015). The high dependence on biomass fuels and presence of brick and sugar industry accentuates the emissions from this region. With annual emissions of 140 Gg, the state of Uttar Pradesh emits the most in IGP followed by West Bengal (57.67 Gg), Bihar (47.8 Gg), Punjab (34.01 Gg), Haryana (26.82 Gg), and the National Capital Territory (NCT) of Delhi (6.74 Gg). The major emissions sources in Uttar Pradesh are kerosene lamps (12%), biomass cooking fuels (30%), brick kilns (20%) and sugar mills (17%). High emissions from IGP and its vicinity to the Himalayas potentially pose a serious threat to water security in the region resulting from impacts to the cryosphere from BC deposition and atmospheric heating.

As discussed in Section 2, best fit probabilistic distributions were obtained for EF and activity data (for each source) using the KS statistic. A sample of 1000 numbers were generated from each of the two distributions (EF and activity data), the product of which provides over one million emission points. Mean and standard deviation was determined for each source using the obtained emission points. The emission points were added up for all the sources to get overall national thousand emission points and subsequently the national mean emission and standard deviation. A best fit probabilistic distribution curve was obtained for the national emission points on the basis of KS statistic. The probabilistic distribution for overall national emissions was found to be General extreme value distribution with KS statistics of 0.01 (Figure 9). Figure 10 presents the sector-wise optimally fit distributions for the BC emissions.

### 4.1   Open Burning

The national level emissions from this sector contribute 12% (103 Gg) to the total emissions. Burning of crop residue has been the major contributor (62%) followed by forest fires (36%). MSW burning contributed only 2% to the open burn-

ing emissions. The source-wise emission contribution and spatially distributed open burning emissions are presented in Figures 7 & 8. The emissions from open burning are highest from the north-west states of Punjab, Haryana (crop residue burning) and north eastern states of Nagaland, Manipur, Mizoram and Tripura (forest fires). Punjab and Haryana are the main food producing states of India, in the months of April, May, October and November, the crop residue is burned for clearing the land for next crop. In the north-east, open burning emissions arise primarily from forest fires, however, some tribal communities also practice slash and burn agriculture in this region as well.

### 4.2  Industry

National level industry sector emissions account for 22% (198 Gg) of the total emissions. In this sector, brick and sugar production contribute the maximum emissions (37% each), followed by steel production (11%), cement (8%) and power plants (7%) (Figure 7). Spatially distributed emissions from the Industry sector are presented in Figure 8. The hotspots of industrial emissions are the states in IGP as most of the brick and sugar industries lie in this area. It is also evident and expected from Figure 8 that metropolitan cities contribute significantly to the sector as they have major industrial belts at the periphery. High emissions from the brick and sugar industry result from the use of low-grade fuels and from dated and inefficient systems and processes. While power plants accounts for 75% of the coal consumption, their BC emissions are just 7% of the total industrial emissions, due to the higher efficiency of combustion in these systems. An acknowledged source of uncertainty in our approach is the lack of specific geolocated coordinate data for the two largest emission sources, brick and sugar.

### 4.3  Transport

Transportation sector emissions account for 17% (154 Gg) of the national BC emissions in 2011. In the transport sector trucks have been found to be most emitting (24%) followed by tractor & trailers (15%). Emissions from bus, car, LCV, LMV and two wheelers contributed 12%, 10%, 13%, 11% and 13% to national transport sector BC emissions respectively. Railways contributed 0.2% to BC emissions in 2011, shipping and aviation combined emitted less than 0.05% (Figure 7). The spatial distribution of Transportation emissions are presented in Figure 8. The main contributors are the metropolitan cities, NCT of Delhi, Mumbai and Bangalore. The results also indicated the majority of the emissions from the transport sector originate from diesel road vehicles (truck, tractors & trailers, bus, LCV and LMV).

### 4.4  Domestic Fuel

Domestic fuels account for almost half of the national BC emissions (47%, 425 Gg). Within the sector, firewood con-

tributes most significantly, (42%), followed by kerosene lamps (26%). Agricultural residue, dung cake, and coal used for cooking contributed 17%, 13% and 2% respectively (Figure 7). Figure 8 shows the spatially distributed emissions of Domestic Fuel usage. Here also, the majority of emissions arise from the IGP due to the high population density in this area. Also the poverty levels are high in this region, so a larger proportion of the population tends to use, cheaper, biofuels for cooking. The biofuel used in hand made stoves has low combustion temperatures leading to inefficient combustion process and consequently the domestic fuel sector has higher BC emissions. Also these are uncontrolled emission sources. Kerosene lamps (109 Gg) are the second highest emitting source as a result of the very high EF of kerosene lamps. While the emissions from kerosene lamps is more than the entire open burning sector combined, studies must be conducted to evaluate the potential impact and transport of this source of BC. It likely has extremely significant health impacts due to the emissions being contained within homes, but the climate impact is likely as large as for Open Burning.

### 4.5  Other

Emissions from this category account for slightly more than 2% (20 Gg) of the national BC emissions. Within this category emissions from use of diesel in irrigation pumps contribute 8 Gg and its use in mobile generators contribute 12 Gg. Among diesel generators their use in mobile towers contribute 4 Gg and other applications (private households, small commercial enterprises and industry) account for remaining 8 Gg.

### 4.6  Uncertainty Analysis

Figure 11 shows the mean and standard deviations based on best fit probabilistic distributions of emissions from the major sectors. Based on the Monte-Carlo simulations using the multiple emissions estimates and available information on uncertainty, the Probability Distribution Functions (PDF) for each of the sectors is calculated as shown in Figure 10. The best fit distribution for the Domestic Fuels sector was found to be Burr distribution with KS statistic of 0.01, for Industrial emissions, Gamma distribution with KS statistics of 0.02, for open burning emissions, Johnson SU distribution with KS statistic of 0.02 and Log-Logistic (3P) for transport sector with KS statistic of 0.03. The uncertainty is highest for emissions from domestic fuels sector. The EFs and activity data for the sources in domestic fuel sector show a large variation leading to high uncertainty in the BC emissions as there is no accurate database of the population using cook stoves, quantity of fuel consumed, and their efficiency.

### 4.7  Comparison with prior estimates

Emissions in this study have been determined using Monte-Carlo simulation of multiple activity data and emission fac-

tors. As previous studies have used point estimates for these highly uncertain quantities, the results are bound to differ. Figure 12 presents the comparison of the results of this study (Table 3) with emission inventories developed in the past. For the base year 2011 the estimate is about 80% than that reported in SAFAR emission inventory (1119 Gg/yr). For inventories with base year 2010, total national emissions estimated in this study are a factor of 1.3 higher than RETRO (697 Gg/yr), factor of 0.8 from than estimated in Klimont et al. (2009) (1104 Gg/yr), factor of 0.9 from that estimated in Lu et al. (2011) (1015 Gg/yr) and were in agreement with emissions determined by Ohara et al. (2007) (862 Gg/yr). All prior national emission estimates lie within two standard deviations of our mean estimate.

Emissions estimates from the Domestic Fuels sector (425±112 Gg/yr) is lower by a factor of 0.7-0.9 with Pandey et al. (2014) (488 Gg/yr), Klimont et al. (2009) (628 Gg/yr) and Lu et al. (2011) (579 Gg/yr). For the Transport sector our emission estimate (154 ± 56Gg/yr) is almost identical to that presented in Sadavarte and Venkataraman (2014) (144 Gg/yr) and a factor of 1.1-1.3 higher than the emissions determined by Lu et al. (2011) (111 Gg/yr), Baidya and Borken-Kleefeld (2009)(123 Gg/yr) and Klimont et al. (2009)(136 Gg/yr). In the Industry sector our emissions (198± 83Gg) are 10-20% lower in view of the inclusion of only higher emitting industries in this study. The combined industrial emission estimate of Sadavarte and Venkataraman (2014) (formal industry) and Pandey et al. (2014) (informal industry) (212 Gg/yr) is in good agreement with our emission estimate (factor 0.94). Estimated industrial emissions are a factor of 0.8-0.9 lower than Lu et al. (2011) (227 Gg/yr) and Klimont et al. (2009) (261 Gg/yr). Emissions from open crop residue burning (64 ± 17Gg/yr) are in close agreement (factor 0.8-1) with Jain (2014)(68 Gg/yr), Lu et al. (2011) (74 Gg/yr) and Pandey et al. (2014) (80 Gg/yr). Forest fire emissions (37 ± 13Gg) are almost identical to those determined in Reddy and Venkataraman (2002a) (39 Gg/yr). As with the national emission estimate, for all sectors prior emission estimates are within one or two standard deviations from our mean emission estimate.

## 4.8 Fuel Balance

A fuel balance approach has been used to ensure that no major emission source has been overlooked in our study. Since biomass consumption data in India is highly uncertain, this approach was only employed for emissions arising from combustion of fossil fuels. Emissions from combustion of diesel, gasoline, fuel oil, ATF, LDO and coal were estimated using emission factors from Streets et al. (2003) and Bond et al. (2004). In 2011, emission from these fuels was estimated to be 281 Gg (Table 4). This was very close to emissions estimated from our methodology (304 Gg) considering the emission sources which use these fuels as a combustion source.

## 4.9 Seasonality of Emissions

There is a strong seasonality associated with BC emissions in India. Crop residue burning, forest fires, brick and sugar industry have a seasonal dependence in emissions. Forest fires are predominant from February to July, monthly BC emissions from forest fires were estimated using MODIS burnt area data. Brick industry becomes active after the monsoon season from months of October to June (Maithel et al., 2012), the sugar industry operates from November to June (Tyagi, 1995) and the emissions are equally distributed among the months of operation. Burning of crop residues generally occurrs in the harvesting months which are October-November for Kharif crop and April-May for Rabi Crop. Emissions of agricultural open burning are equally distributed among the months of April, May, October and November. For all the other sources, emission rates are assumed to be uniform throughout the year. Using this data monthly variation of BC emissions has been estimated and is shown in Figure 13.

The emissions in April are highest due to the burning of crop residues. Despite the absence of crop residue burning, emissions in March are also high because of the emissions from forest fires. As we have shown a considerably amount of the emissions to come from the IGP in close proximity to the Himalaya, this causes further concern to the potential cryospheric impact of these aerosols as they are strongest during the period when the seasonal snow melt period is beginning and could be incorporated into the snow pack.

## 5 Conclusions

A spatially resolved Black Carbon (BC) emission inventory for 2011 has been developed considering major sectors with careful consideration for sub-sector sources. The sources were classified into five major sectors: i) Open Burning, including: forest fire emissions, open solid waste burning, and agriculture residue burning; ii) Industry, including: brick industry, cement, steel plants, sugar mills and power plants; iii) Transport, including: two wheelers, cars, light motor vehicles passenger, light commercial vehicles, taxies, trucks, buses, tractors & trailers, railways, shipping and airways; iv) Domestic Fuel, including: firewood burning, agricultural residue, coal, liquid petroleum gas, kerosene (cooking and lighting) and dung cake; and v) Other, including: use of diesel in irrigation pumps and for other power generation in diesel generators.

This is a first-of-kind comprehensive study which included sources such as kerosene lamps and forest fires that were not part of earlier emission inventories. Furthermore, for each sector, source uncertainties in emissions have been estimated based on variability in available activity data and emission factors. Lastly, and significantly, we provide our estimate of emissions at monthly temporal resolution on a spatially distributed 40 x 40 km. grid.

The national BC emissions for India in 2011 are estimated to be 901 ± 152 Gg/yr with domestic fuels contributing maximum (47%) followed by industries (22%), transport (17%), open burning (12%) and others (2%). Large emission in domestic fuels sector stems from the extensive use of biomass for cooking in India. Firewood is the single largest emitter with 177 Gg (20%) of BC emissions in 2011. The emissions from firewood are more than the entire transportation sector combined. Kerosene lamps surprisingly contribute 12% to the national BC emissions. The emissions have been found to be have a significant seasonality, varying from 55 Gg in July to 90 Gg in April 2011.

The results of the study could be used to assess the contribution of different sources to national and regional emissions. The spatial resolution of the inventory should be useful for modelling the Black Carbon processes in atmosphere through air quality models. Monthly gridded emission datasets can also be prepared for finer temporal resolution input. To improve the future BC emission estimates, local emission factors and activity data should be improved, especially for domestic fuels and brick industry. Emission inventory can be improved nationally, regionally and temporally by comparing the modeled emission estimates (providing the inventory as input to air quality models) with the observed data.

*Acknowledgement.* This work was conducted within the Norwegian Research Council's INDNOR: Hydrologic sensitivity to Cryosphere-Aerosol interaction in Mountain Processes (HyCAMP) (Researcher project - MILJØ2015 #222195) and The Department of Science and Technology, Government of India through Grant no. INT/NOR/RCN/P-05/2013. We are grateful for constructive feedback received from two anonymous reviewers and our editor who encouraged the addition of the road network and incorporating the fuel balance analysis.

*Data Availability* The inventory is available through request of the authors. More information may be found at: http://www.mn.uio.no/geo/english/research/projects/hycamp

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

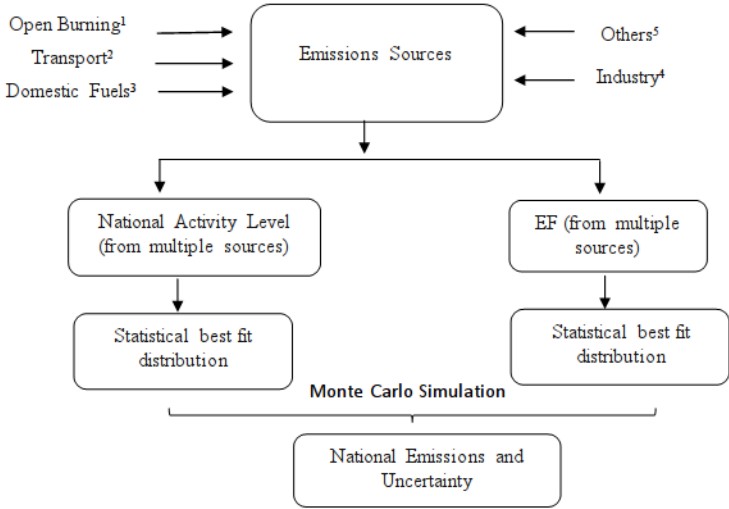

**Figure 1.** Methodology for National Emissions.

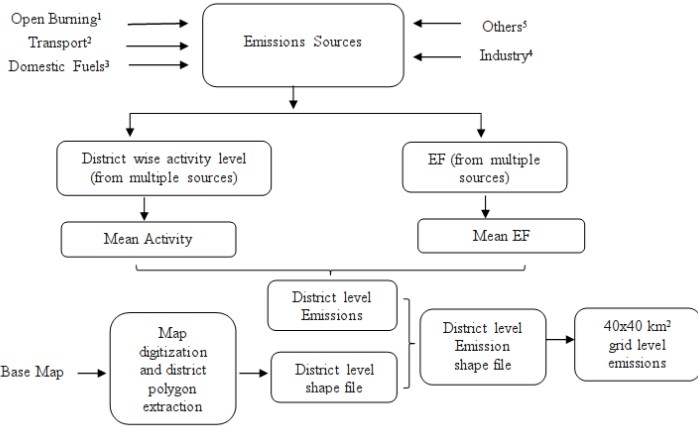

**Figure 2.** Methodology for preparing gridded emissions.

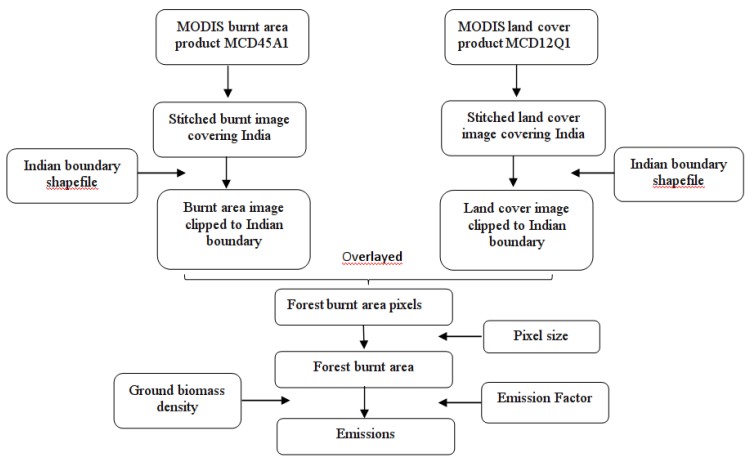

**Figure 3.** Flow chart for the calculation of Forest Fire emissions based on MODIS products.

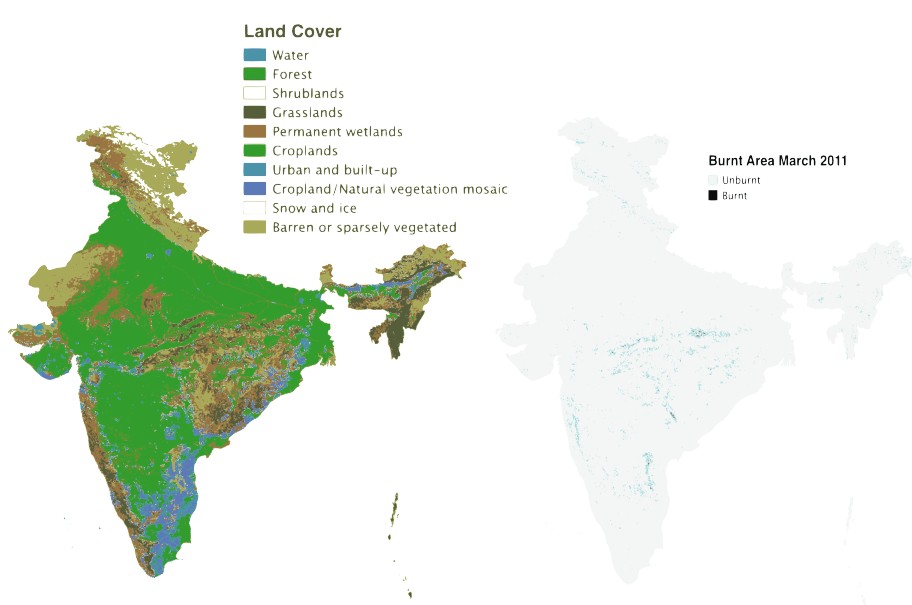

**Figure 4.** Land Cover and Burnt Area for March 2011.

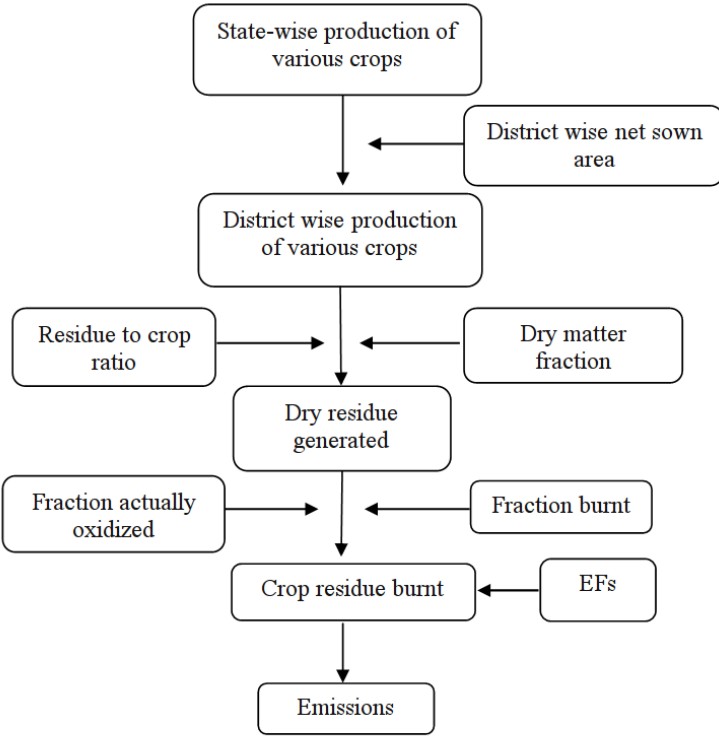

**Figure 5.** Flow chart for the calculation of agricultural waste burning.

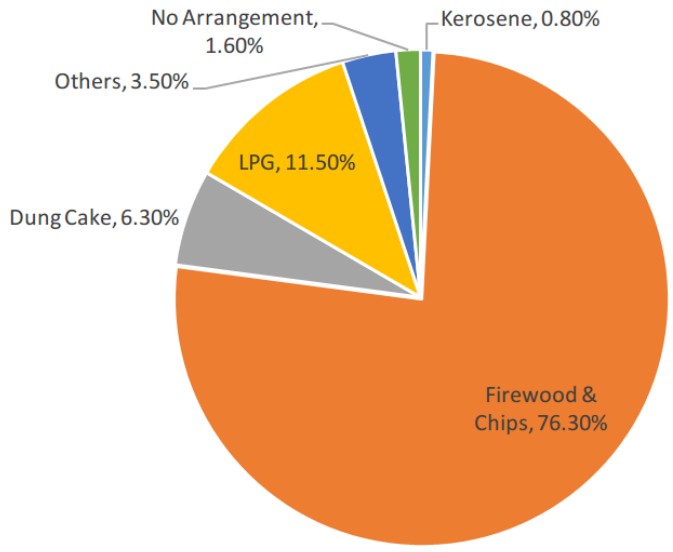

**Figure 6.** Energy sources used for cooking in Rural India 2009-2010.

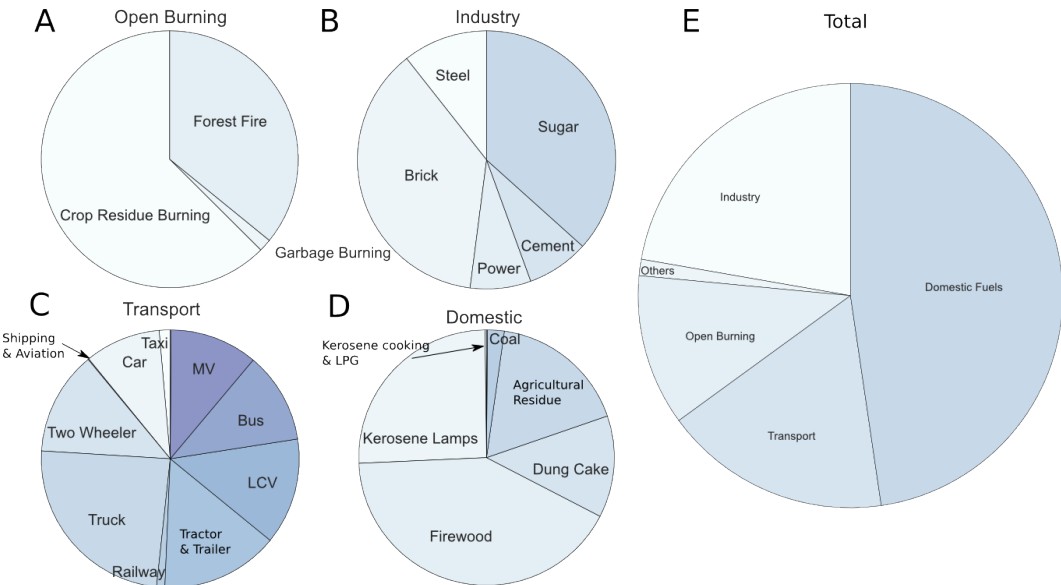

**Figure 7.** (a-d) Proportion of sub-categories to the major sector emissions and e) contribution of major sector emissions to the national emissions total.

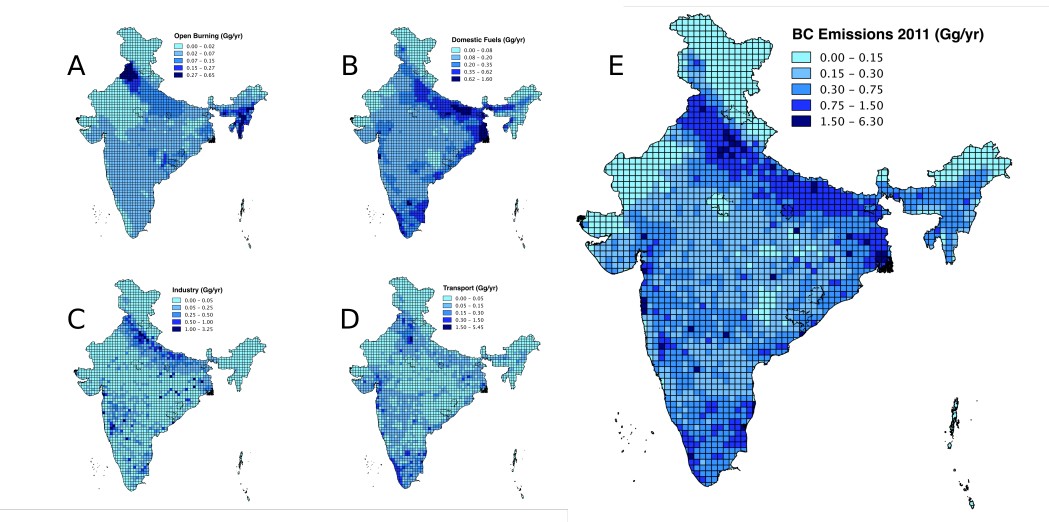

**Figure 8.** (a-d) Maps of major sector emissions and e) spatial variability of national emissions total for BC.

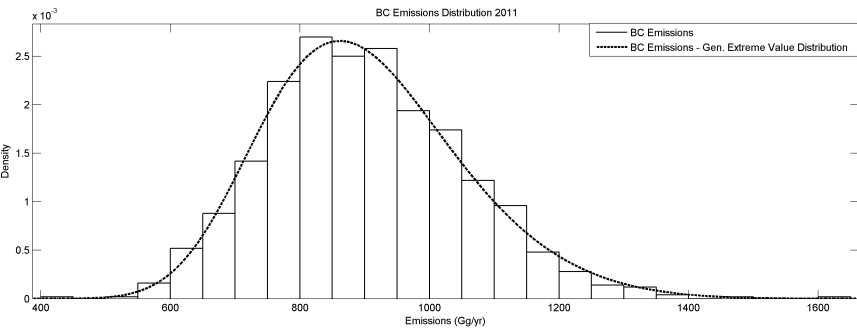

**Figure 9.** Gen. Extreme Value Distribution fit for the National BC Emissions.

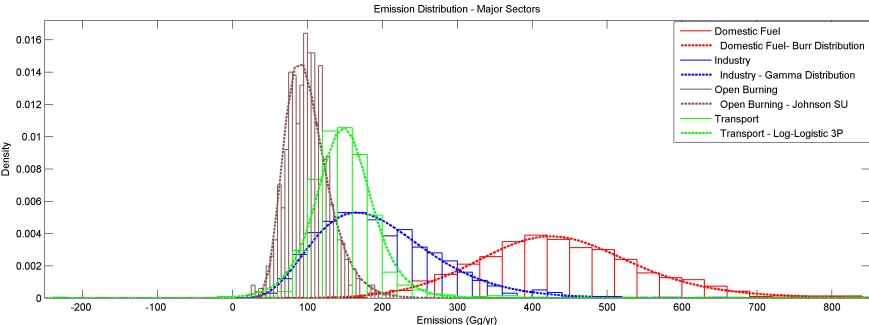

**Figure 10.** Sectorial emission histograms and associated best-fit PDFs.

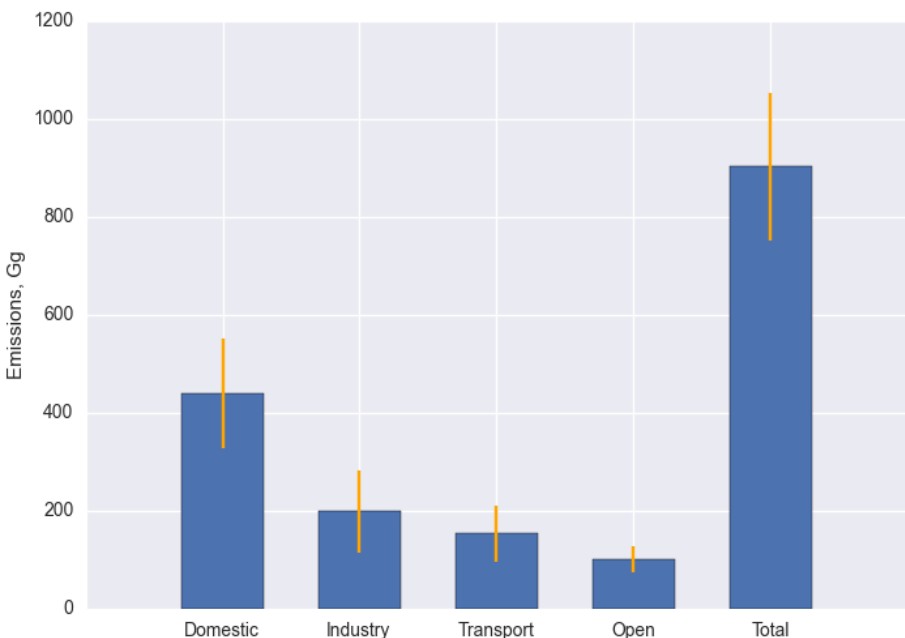

**Figure 11.** Mean and Standard Deviation for each of the major sectors of emissions for India, 2011.

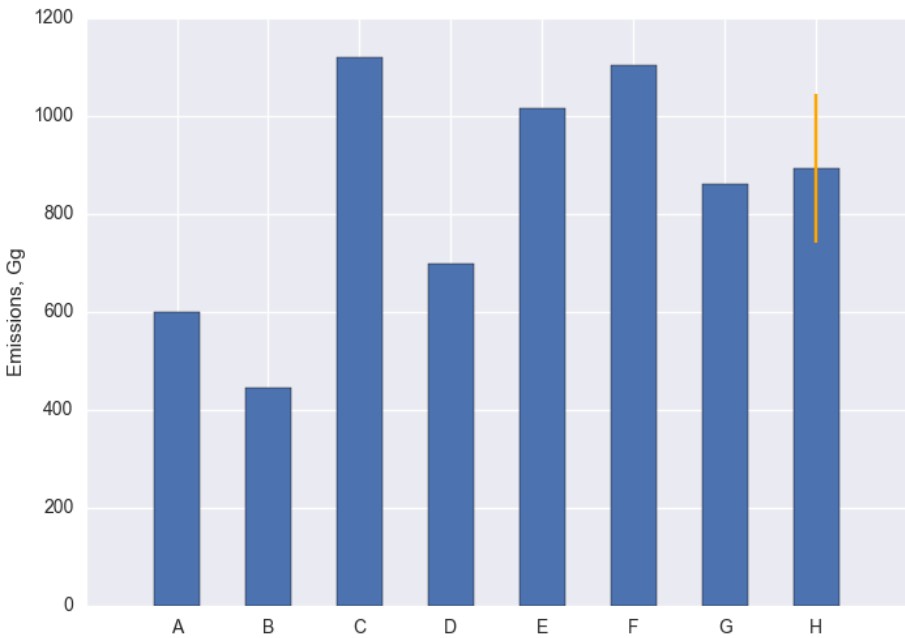

**Figure 12.** Comparison of current BC emissions estimate with prior published results for India. A: Streets et. al. 2003 (Base year 2000); B: Reddy and Venkataraman 2002a & 2002b (Base year 1997); C: Sahu, et. al. 2008 (Base year2011); D: Schultz et. al. 2008 (Base year 2010); E: Lu et. al.(2011) (Base year 2010); F: Klimont et. al.(2009) (Base year 2010); G: Ohara et. al.(2007) (Base year 2010); H:This study (Base year 2011)

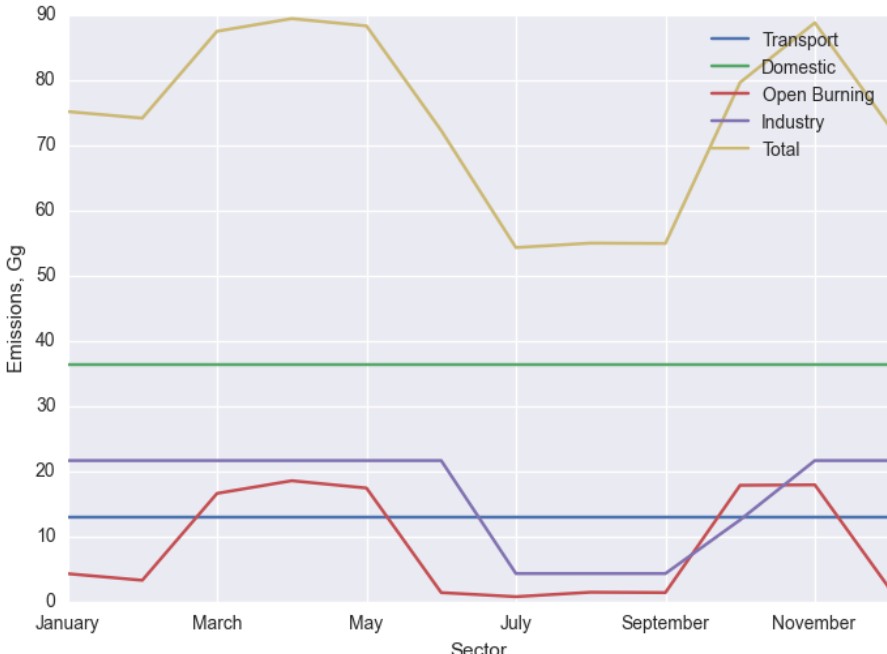

**Figure 13.** Timeseries of monthly emissions for India, 2011. Note the strong seaonality of the Open burning and Industry sector. In the latter case, the seasonality results predominately from the sugar cane production.

**Table 1.** Mean activity data, standard deviation, and best fit probabilistic distribution.

| Subsector | Activity Level | Mean ± SD | Distribution |
|---|---|---|---|
| **Open Burning**($MT/year$) | | | |
| Crop Residue Burning | 99.93[1,2], 89.79[1,3], 90.94[1,4] | 93.56±4.96 | Uniform |
| Forest Fire* | 47.83[5] | 47.83±9.56 | Normal |
| Garbage Burning | 3.90[2,6], 2.51[2,6,7,8,9] | 3.2±0.76 | Uniform |
| **Industry** ($MT/yr$) | | | |
| Brick* | 474[10] | 474±237 | Normal |
| Steel* | 40.05[11] | 40.05±8.01 | Normal |
| Sugar* | 77.1[12,13] | 77.1±15.42 | Normal |
| Cement* | 28.06[14] | 28.06±5.61 | Normal |
| Power Coal* | 380.91[15] | 380.91±76.18 | Normal |
| Power Diesel* | 0.71[15] | 0.71±0.01 | Normal |
| **Transport** ($BillionKm/yr$) | | | |
| Bus | 38.77[16,17], 39.3916[16,18], 32.8316[16,19] | 35.46±3.94 | Uniform |
| Car | 128[16,17], 196.27[16,18], 130.85[16,19], 167.87[16,21] | 155.06±29.76 | Uniform |
| LMV | 104.22[16,17], 131.51[16,18], 65.75[16,19] | 105.32±22.74 | Uniform |
| LCV | 78.55[16,17], 99.12[16,18], 74.34[16,19] | 111.37±38.70 | Uniform |
| Truck | 122.99[16,17], 87.30[16,20], 125.50[16,21] | 109.60±16.72 | Uniform |
| Taxi | 38.25[16,17], 40.26[16,18], 48.32[16,19], 16.91[16,21], 60.44[16,22], 51.01[16,23] | 42.52±14.86 | Gumbel |
| Two Wheeler | 450.8[16,17], 764.07[16,18], 481.36[16,21], 2062.89[16,22], 313.26[16,23] | 814.50±716.18 | Uniform |
| Tractor & Trailer | 11.08[16,17], 26.38[16,18] | 18.73±8.38 | Uniform |
| Railway Coal (KT/yr)* | 1[24] | 1±0.02 | Normal |
| Railway Diesel (KT/yr)* | 21.06[24] | 21.06±42.10 | Normal |
| Shipping HSDO (KT/yr)* | 0.11[25,26] | 0.11±0.02 | Normal |
| Shipping Fuel Oil (KT/yr)* | 80[25,26] | 80±16 | Normal |
| Shipping LDO (KT/yr)* | 0.36[25,26] | 0.36±0.07 | Normal |
| Aviation LTO (KT/yr)* | 514.16[2,25,27,28] | 514.16±102.83 | Normal |
| Aviation Cruise (KT/yr)* | 1505.83[2,25,27,28] | 1505.83±301.16 | Normal |
| **Domestic Fuel** ($MT/yr$) | | | |
| Dung Cake | 144.84[29], 75.62[30] | 110.23 ± 37.91 | Uniform |
| Agriculture Residue | 125.34[29], 81.25[30] | 103.30±24.14 | Uniform |
| Firewood | 209.99[31], 281.99[29], 193.87[30] | 228.62±41.96 | Uniform |
| Coal* | 4.77[31] | 4.77±0.95 | Normal |
| Kerosene Cooking* | 4.57[31,32] | 4.57±0.91 | Normal |
| LPG* | 12.37[31] | 12.37±2.47 | Normal |
| Kerosene Lamps | 1.68[32], 1.21[31,32] | 1.45±0.25 | Uniform |
| **Others** ($MT/yr$) | | | |
| Irrigation Pumps* | 2.11[25] | 2.11±0.42 | Normal |
| Diesel Generators(Mobile Towers)* | 1.12[25,33] | 1.12±0.22 | Normal |
| Diesel Generators(Other)* | 2.28[25,33] | 2.28±0.45 | Normal |
| **Sources** | [1]Ministry of Agriculture (2013) [2]IPCC 2006 (2006)[3]Jain (2014)[4]Venkataraman et al. (2005)[5]Land Processes Distributed Active Archive Center (LP DAAC), 2000[6]CPCB (2007)[7]Kumar (2010)[8]National Environmental Engineering Research Institute (NEERI) (2010)[9]CPCB (2012) [10]Industry experts, [11]Press Information Bureau (2011)[12]DAC (2013)[13]ISMA (2012)[14]CMA[15]CEA (2012)[16]Ministry of Road Transport and Highways (2011)[17]Baidya and Borken-Kleefeld (2009)[18]Ramachandra et al. (2015)[19]Guttikunda and Calori (2013)[20]Mittal and Sharma (2003)[21]Ramachandra and Shwetmala (2009)[22]Sindhwani and Goyal (2014)[23]Pandey and Venkataraman (2014)[24]Ministry of Railways (2012b)[25]MoPNG (2013), [26]EEA (2013)[27]ICAO (2010)[28]DGCA (2013)[29]Yevich (2003)[30]Smith et al. (2000)[31]MoSPI (2014b)[32]Lam et al. (2012)[33]Shakti Sustainable Energy Foundation (2014)* Normal Distribution Assumed | | |

**Table 2.** Mean EFs, standard deviation and best fit probabilistic distribution.

| Subsector | EFs Used | Mean EF ± SD | Best Fit Distribution |
|---|---|---|---|
| **Open Burning**($g/kg$) | | | |
| Crop Residue Burning | 0.69[1], 0.78[2], 0.73[3], 0.47[4], 0.75[2] | 0.69±0.19 | Dagum |
| Forest Fire | 0.56[1], 0.98[4], 0.99[5], 0.56[6] | 0.76±0.21 | Error |
| Garbage Burning | 0.65[7], 0.37[8] | 0.51±0.15 | Uniform |
| **Industry**($g/kg$) | | | |
| Brick | 0.11[9], 0.27[9], 0.09[9] | 0.16±0.09 | Uniform |
| Steel | 0.32[3], 1.1-1.58[10], 0.224[11], 0.23-0.13[12], 0.06[5], 0.0095[13] | 0.45±0.51 | Log Pearson 3 |
| Sugar | 1.2[14], 0.7[15] | 0.95±0.27 | Uniform |
| Cement | 0.32[3], 1.1-1.58[10], 0.224[11], 0.23-0.13[12], 0.06[5], 0.0095[13] | 0.45±0.51 | Log Pearson 3 |
| Power Coal | 0.003-0.032[16], 0.077[13], 0.0029[11], 0.002[5], 0.06[5] | 0.03±0.03 | Gamma (3P) |
| Power Diesel | 0.25[11], 0.15[8], 0.06[13] | 0.15±0.08 | Uniform |
| **Transport**($g/km$) | | | |
| Bus | 0.35[17,18], 0.8[18,19], 0.225[18,20], 0.61[18,21] | 0.49 ± 0.24 | Uniform |
| Car | 0.16[22], 0.17[17,18], 0.05[18,19], 0.07[18,20], 0.16[18,21] | 0.09±0.06 | Uniform |
| LMV | 0.16[22], 0.138[17,18], 0.17[18,21] | 0.15±0.01 | Uniform |
| LCV | 0.27[17,18], 0.13[18,19], 0.16[18,21] | 0.19 ±0.07 | Uniform |
| Truck | 0.61[17,18], 0.26[18,19], 0.19[18,20], 0.31[18,21] | 0.34±0.17 | Uniform |
| Taxi | 0.01[22], 0.06[17,18], 0.076[18,20], 0.057[18,21] | 0.05±0.03 | Uniform |
| Two Wheeler | 0.013[23], 0.012[17,18], 0.038[18,19], 0.023[18,20] | 0.02±0.01 | Uniform |
| Tractor & Trailer* | 1.24[23] | 1.24 ±0.25 | Normal |
| Railway Coal (g/Kg) | 1.83[13], 3[8] | 2.415±0.33 | Uniform |
| Railway Diesel (g/Kg) | 1.53[24], 0.51[8], 0.29[13] | 0.78±0.59 | Uniform |
| Shipping HSDO (g/Kg) | 0.85[25], 1.19[8], 1.32[26], 0.36[25] | 0.78±0.49 | Gen. Extreme Value |
| Shipping Fuel Oil (g/Kg) | 0.38[25], 0.36[25], 0.97[25], 0.85[25], 1.19[8], 1.32[26] | 0.72±0.40 | Wakeby |
| Shipping LDO (g/Kg) | 0.85[25], 1.19[8], 1.32[26] | 0.89±0.46 | Uniform |
| Aviation LTO (g/Kg) | 0.08-0.1[27] | 0.09±0.01 | Uniform |
| Aviation Cruise (g/Kg) | 0.02-0.1[27] | 0.06±0.02 | Uniform |
| **Domestic Fuel** ($g/kg$) | | | |
| Dung Cake | 0.53[8], 1[14], 0.8[28], 0.25[4], 0.49[29], 0.18[30], 0.12[31], 0.4[15] | 0.47 ± 0.31 | Gen. Extreme Value |
| Agriculture Residue | 0.43[32], 0.66[11], 0.75[2], 0.47[4], 0.37[29], 1[8], 1.3[33], 0.24[30], 1.38[34], 0.6[31], 0.9[31] | 0.74 ± 0.37 | Gen. Extreme Value |
| Firewood | 1[3], 0.59[1], 0.41[4], 0.7[32], 1.2[14], 1[28], 0.85[8], 0.6[31], 0.35[29], 1.1[33], 0.25[30], 0.83[35], 1.33[6], 0.7[36] | 0.78 ± 0.32 | Gen. Extreme Value |
| Coal | 1.91[3], 2.84[10], 1.83[4], 5[8], 0.28[37], 2.295[24], 0.8[11], 0.3[11], 0.69[11], 0.79[11], 0.32[11], 0.497[11], 0.07[36], 5.4[15] | 1.64 ± 1.73 | Pearson 6 (4P) |
| Kerosene Cooking | 0.16[4], 0.02[15] | 0.18 ± 0.02 | Uniform |
| LPG | 0.067[11], 0.01[15] | 0.04 ± 0.03 | Uniform |
| Kerosene Lamps | 66[38], 89[38], 72[38], 110[38], 79[38], 94[38], 89[38], 76[38] | 84.37 ± 14.05 | Pearson 6 (4P) |
| **Others** ($g/kg$) | | | |
| Irrigation Pumps | 3.18[24], 3.96[8] | 3.56±0.22 | Uniform |
| Diesel Generators | 3.41[24], 3.96[8] | 3.68±0.16 | Uniform |
| **Sources** | [1]Andreae and Merlet (2001)[2]Turn et al. (1997)[3]Streets et al. (2001)[4]Reddy and Venkataraman (2002a)[5]Qin and Xie (2011)[6]Zhang et al. (2013)[7]Akagi et al. (2011)[8]Bond et al. (2004)[9]Weyant et al. (2014)[10]Cooke et al. (1999)[11]Ni et al. (2014)[12]Novakov (2003)[13]Reddy and Venkataraman (2002b)[14]Liousse et al. (1996)[15]Pandey et al. (2014)[16]Streets et al. (2003)[17]ARAI (2008)[18]Chow et al. (2011)[19]Borken et al. (2008)[20]Baidya and Borken-Kleefeld (2009)[21]Mittal and Sharma (2003)[22]Reynolds and Kandlikar (2008)[23]TERI (The Energy and Resources Institute) (2006) [24]Ito and Penner (2005)[25]Lack et al. (2009)[26]Bond et al. (2007)[27]Hendricks et al. (2004)[28]Cachier (1998)[29]Saud et al. (2012)[30]Sen et al. (2014)[31]Habib et al. (2004)[32]Li et al. (2009)[33]Parashar et al. (2005)[34]Shen et al. (2010)[35]Shen et al. (2012)[36]Chen et al. (2009)[37]Chen et al. (2005)[38]Lam et al. (2012)* Normal Distribution Assumed |

**Table 3.** Mean national emissions and standard Deviation.

| Sector/Subsector | Emissions (Gg/yr) |
|---|---|
| **Open Burning** | **102.84 ± 27.56** |
| Crop Residue Burning | 64.31 ± 17.19 |
| Forest Fire | 36.90±12.85 |
| Garbage Burning | 1.63 ± 0.62 |
| **Industry** | **198.5 ± 83.391** |
| Brick | 74.11 ± 61.38 |
| Steel | 21.09 ± 32.18 |
| Sugar | 72.76 ± 25.05 |
| Cement | 15.45 ± 22.26 |
| Power | 15.09±23.88 |
| **Transport** | **154.34 ± 56.14** |
| Bus | 17.64 ± 8.72 |
| Car | 14.69 ± 10.54 |
| LMV | 17.01 ± 25.03 |
| LCV | 20.62 ± 10.51 |
| Truck | 37.46 ± 20.49 |
| Taxi | 2.13 ± 1.44 |
| Two Wheeler | 20.11 ± 39.50 |
| Tractor & Trailer | 22.79 ± 11.41 |
| Railway | 1.60 ± 1.32 |
| Shipping | 0.15± 0.07 |
| Aviation | 0.14±0.04 |
| **Domestic Fuel** | **425.36 ± 111.97** |
| Dung Cake | 54.79 ± 48.15 |
| Agriculture Residue | 74.38 ± 44.17 |
| Firewood | 177.34 ± 83.88 |
| Coal | 9.02 ± 14.622 |
| Kerosene Cooking | 0.83 ± 0.19352 |
| LPG | 0.47 ± 0.39 |
| Kerosene Lamps | 108.53 ± 27.10 |
| **Others** | **20.08 ± 2.59** |
| Irrigation Pumps | 7.55 ± 1.73 |
| Diesel Generators(Mobile Towers) | 4.14 ± 0.85 |
| Diesel Generators(Other) | 8.39 ± 1.73 |
| **Total** | **901.11±151.56** |

**Table 4.** Fuel Balance.

| Sector/Fuel | Activity (MT) | EF (g/Kg) | Emission (Gg) |
|---|---|---|---|
| Coal | 535.88[1] | 0.328[2] | 175.77 |
| Gasoline/Petrol | 14.442[3] | 2.795 [4] | 40.37 |
| Diesel | 63.504[3] | 1.02[4] | 64.77 |
| Fuel Oil | 6.624[3] | 1.02[4] | 64.77 |
| ATF | 5.324[3] | 0.03[4] | 64.77 |
| **Total** | | | **281.33** |

**Sources**   [1]MoSPI (2014a)[2]Streets et al. (2003)[3]MoPNG (2014)[4]Bond et al. (2004)