# Peer review of "Monthly and Spatially Resolved Black Carbon Emission Inventory of India: Uncertainty Analysis"

_Atmospheric Chemistry and Physics, 2015_

## Referee Comment (RC1) · Anonymous Referee #1 · 23 Mar 2016

These are few raised concern by referee : 1) To build a scientifically viable BC emission inventory for developing country like India is very curial and sensitive for climate scientific point of view. I am happy that author has adopted some approach to build an inventory but same time I am not agreed with the methodology and approached they adopted. Moreover the spatial data used to build such a fine resolution inventory is not up to mark and nearly impossible. To use the inventory in atmospheric model, it needs to be scientifically valid rather than just to adopt a statistical approach to come up an inventory. 2) The title of manuscript clams its spatially resolved BC emission inventory of India But in reality there is very little or very poor spatial/geographical data are being used to build a 40x40 km resolution inventory. In fact there is no road network data is used or any discussion. If you input emission estimations are at such a coarse level i.e. district and state level then how can you generated spatially resolved inven-

tory at such fine resolution? It is very easy to interpolate in GIS environment to any finer resolution. This kind of inventory will mislead in further application is in climate or chemistry model. It is likely to introduce further uncertainty in terms of spatial allocation. 3) I would like to suggest author to enlighten the detail of industrial units/ activity data used/its spatial pattern over India for present work. There is little discussion about activity data for each sector. I will suggest elaborating each dataset quantitatively. 4) There is no road network data used to allocate transport emissions is another big gap in developing robust and reliable inventory. 5) Some sectors like Mobile towers, etc. are calculated at very course resolution like state level. 6) An inventory has to build based appreciate emission factor which is scientifically correct and suitable to Indian condition rather than just emission factor/activity data generated based on statistical approach. An emission inventory like climate agent BC has to be scientifically correct in terms of selection of emission factors data due to existing large uncertainty of the order of 2 to 4. We have to very careful in selecting the EFs as most of available EFs are generated for developed countries except in few cases. It is extremely important to select an EF suit to Indian condition rather than taking EF derived from simple statistical approach. This is a ongoing big issue and major reason of debate among scientific community across the globe. It has to be tackle scientifically instead of statistically. 7) Emission inventory are never being interpolated with coarse level emission estimation i.e. state and district level data. I can agree for state/district level national estimation but can not with 40x40km surface emission date. 8) I am very surprised that NCT contributes just 6.7 Gg/yr of BC but in reality NCT account nearly between 5-10 % of transport and industrial load in India with high population density. This number appears to be very small and unrealistic. I will recommend author to recalculation of emission especially sensitive regions like NCT and other industrial regions. These regions are very sensitive due to air quality issues in recent times. I pretty sure the BC estimation over NCT should have been much higher than author's calculation. Otherwise this kind of miss representation may be due to unappreciated emission factor and approach used for estimation.

---

## Referee Comment (RC2) · Anonymous Referee #2 · 23 Mar 2016

I have only good words to say about this article. It is one of the easiest articles I have ever read and it flows very well for the reader. The science behind is novel and it would be a great tool for any modeller wants to include emission data from India to any transport, chemistry or climate model.

1) I am sceptic about the methodology used to create this spatial resolution, although the input was at a district level, hence, in course resolution. However, if we exclude inverse modelling approaches that would be able to create a fully resolved inventory, I admit this approach is the best we have.

2) Another problem is the emission factors used in the study. I expect that the authors will justify more their choice of emission factor selection, since there are already-known and validated emission factors in the literature used for such purposes (showed also in

Table 2). I am not sure if taking the mean emission factor would make sense since in some cases it varies a lot. In addition, I do not think that emission factors taken from Andrae and Merlet's paper (used in most models) can be given the same weight with other emission factor studies. Of course, since they do not validate their dataset, they do not need to evaluate the emission factors they used.

3) I think that a missing point is evaluation of the database created from this work. I would highly recommend to the authors to try including this dataset to a global inventory and perform a few runs comparing the surface concentrations of BC from their model and the dataset with observations in Southeastern Asia. There are several measuring stations there and such a comparison would help all of us understand how valid this dataset is. At the moment, a poor discussion is performed for such an interesting topic. I believe that including the dataset to a model and performing the aforementioned analyses would expand the discussion a lot.

4) My highest concern is if the present manuscript falls within the scope of the journal. According to the main page of the journal, it is "dedicated to the publication and public discussion of studies investigating the Earth's atmosphere and the underlying chemical and physical processes in an altitude range from the land and ocean surface up to the turbopause, including the troposphere, stratosphere, and mesosphere. The main subject areas comprise atmospheric modelling, field measurements, remote sensing, and laboratory studies of gases, aerosols, clouds and precipitation, isotopes, radiation, dynamics, biosphere interactions, and hydrosphere interactions. The journal scope is focused on studies with general implications for atmospheric science rather than investigations that are primarily of local or technical interest. In my opinion, the present study comprises a rather statistical methodology than a modelling, measurements, lab-based methodology (perhaps a bit of GIS) that the journal requires. Furthermore, without the evaluation using model-simulated concentrations and comparison with measurements of the Southeastern Asia, the study is of local interest.

A general comment is that the Editor should have probably rejected the present

manuscript in its current form. I have seen several more subject-related manuscripts being rejected. However, I am glad he did not and the authors have the chance to provide a very useful database to the modelling community (after evaluation of their results, of course).

––––––––––––––––––––––––––––––––

---

## Author Comment (AC1) · 27 May 2016

**Abstract**

We have received and are grateful for the response provided by the two anonymous reviewers of our manuscript. The comments show a detailed and good understanding of the intention of the dataset and ambitions that we have for the scientific community. Further, they show a careful consideration of the data we used and amount of work that has gone into this 'bottom-up' analysis. Individual comments are addressed separately in the following sections. Original comments are in bold, with our responses following.

[Figure]

**1 Reviewer #1 comments**

AR-1 raised very good concerns, many of which are common with our own. We feel that the biggest challenge with providing accurate assessments of the climatologically significant black carbon aerosol arises from having an accurate estimate of the emissions. We have provided our emission inventory as an attempt to provide a detailed inventory that provides both temporal variability and a good statistical evaluation of the uncertainty with the different sources.

**These are few raised concern by referee:**

**1) To build a scientifically viable BC emission inventory for developing country like India is very curial and sensitive for climate scientific point of view. I am happy that author has adopted some approach to build an inventory but same time I am not agreed with the methodology and approached they adopted. Moreover the spatial data used to build such a fine resolution inventory is not up to mark and nearly impossible. To use the inventory in atmospheric model, it needs to be scientifically valid rather than just to adopt a statistical approach to come up an inventory.**

We recognize that methodologically there are several different approaches that may be utilized for the development of an emission inventory. However, the reviewer provides no example of alternative methods they would recommend. It seems the primary concern of the approach employed in our work is related to the 40 km. resolution that we have used to prepare the data. There are inverse methods that could be employed to address these concerns, but beyond the fact that such an analysis would be beyond the scope of this work, some of the methods require a good 'bottom-up' inventory as a starting point (e.g. Wang et al., 2016). We feel this is precisely where our work contributes, and are enthusiastic for inclusion in such an analysis. Further, we are aware of other recent work to assess the quality of emission inventories using inverse modeling methods (e.g. Fang et al., 2014; Thompson and Stohl, 2014.), but note that these are

generally limited to species for which "the atmospheric loss (if any) can be described as a linear process and can be used on continental to regional and even local scales with little or no modification." Thompson and Stohl (2014). For black carbon this assumption would be invalid.

**2) The title of manuscript clams its spatially resolved BC emission inventory of India But in reality there is very little or very poor spatial/geographical data are being used to build a 40x40 km resolution inventory. In fact there is no road network data is used or any discussion. If you input emission estimations are at such a coarse level i.e. district and state level then how can you generated spatially resolved inventory at such fine resolution? It is very easy to interpolate in GIS environment to any finer resolution. This kind of inventory will mislead in further application is in climate or chemistry model. It is likely to introduce further uncertainty in terms of spatial allocation.**

We are glad that the reviewers raised this issue. The uniform distribution of emissions in a district from point sources would have definitely reduced the accuracy of our dataset. We have now changed the method and a considerable effort was undertaken to resolve this concern by using the precise location coordinates of industrial structures and locating them in a grid element. This way the emissions from a point source would only be highlighted in a particular grid as compared to uniform distribution in the complete district previously. We have done this analysis for all the sources in Industry category (Steel Plants, Cement Plants, Coal & Diesel Power Plants and Sugar Mills) except Brick Industry for which little or no data is available regarding their location.

We also found the reviewers suggestion regarding the use of road network data constructive, and have now used this information in distributing emissions from districts to grids. Road network data from Open street maps has been used for this process. The emissions were distributed from districts to grids according the proportion of road length of the district contained in the grid. In the case of a grid lying in multiple districts the emission in each segment of grid was estimated according to the aforementioned

methodology and emissions from all these segments were added to get emission in a grid. Text indicating the changes in method have been included in Sections 2 & 3. We believe this change will definitely improve the quality of our dataset and thank reviewers for raising the concern.

**3) I would like to suggest author to enlighten the detail of industrial units/ activity data used/its spatial pattern over India for present work. There is little discussion about activity data for each sector. I will suggest elaborating each dataset quantitatively.**

We were reluctant to include substantial information regarding the activity data in order to keep the manuscript succinct. However, in our revisions we have tried to appropriately detail each activity dataset further in the new draft of the manuscript.

**4) There is no road network data used to allocate transport emissions is another big gap in developing robust and reliable inventory.**

Accepted. Kindly refer Point 2.

**5) Some sectors like Mobile towers, etc. are calculated at very course resolution like state level.**

The reviewer is correct that the initial data used in our inventory for some emissions is from state-level reports or available data sets. However, for none of the sources did we consider the emissions only at the state-level. In the case of mobile-towers or municipal waste burning, we used district-level population data to distribute the emissions to a finer resolution. For agricultural burning, we used the district-level reporting of net sown area. We are confident that this is a reasonable method to bring greater granularity to the data and distribute it to a higher resolution.

Lack of data was a serious challenge in the development of this inventory. For instance, India has around half a million mobile towers with information regarding their location only available at state level. We wanted to make the inventory as comprehensive as

possible by considering all the significant sources. In view of lack of data we had to use suitable proxies to get data as a finer resolution. Also note that, Mobile towers contributes less than 0.5% to the national emissions and so would not have significant impact on the overall magnitude of emissions.

**6) An inventory has to build based appreciate emission factor which is scientifically correct and suitable to Indian condition rather than just emission factor/activity data generated based on statistical approach. An emission inventory like climate agent BC has to be scientifically correct in terms of selection of emission factors data due to existing large uncertainty of the order of 2 to 4. We have to very careful in selecting the EFs as most of available EFs are generated for developed countries except in few cases. It is extremely important to select an EF suit to Indian condition rather than taking EF derived from simple statistical approach. This is a ongoing big issue and major reason of debate among scientific community across the globe. It has to be tackle scientifically instead of statistically.**

We recognize that the lack of country specific emission factors is a big challenge in development of an accurate emission inventory and we share this concern. However, for a country like India with such a vast geography and variation in terms of technologies used for combustion processes even a country specific single emission factor would not do justice. For instance, considering firewood combustion as a domestic fuel, each community uses a different type of cooking stove and likely even different types of wood. With the difference in structure of the combustion technology, emission factors are bound to vary and a single emission factor would not be sufficient to address the variation. This is exactly the issue we aimed to resolve through the statistical approach to emission inventories which enables us to better preserve and quantify the uncertainty. We wish to assess the uncertainty in the emissions in India because of these variation in technologies and the paucity of data associated to them.

We absolutely agree that it would be incorrect to use the emission factors derived

for developed countries for emission estimation in India. In view of this, most of our emission factors are from studies conducted in India , China or South Asia. In cases where we have derived factors from the global studies Bond (2004); Streets (2003); Akagi et al. (2011); Ito and Penner (2005) etc.) mostly we have used the emission factors explicitly given for India/South Asia/Developing country.

**7) Emission inventory are never being interpolated with coarse level emission estimation i.e. state and district level data. I can agree for state/district level national estimation but can not with 40x40km surface emission date.**

We have used strong and best available proxies to resolve data to a finer resolution(Population for domestic fuels, net sown area for agricultural residue burning) which should provide a fair estimate of emissions. Further, the changes that we have made in our methodology on the recommendation of both reviewers (inclusion of road network data, treating industrial units as point source) improves the accuracy of our 40x40 km2 spatially resolved dataset.

**8) I am very surprised that NCT contributes just 6.7 Gg/yr of BC but in reality NCT account nearly between 5-10 % of transport and industrial load in India with high population density. This number appears to be very small and unrealistic. I will recommend author to recalculation of emission especially sensitive regions like NCT and other industrial regions. These regions are very sensitive due to air quality issues in recent times. I pretty sure the BC estimation over NCT should have been much higher than author's calculation. Otherwise this kind of miss representation may be due to unappreciated emission factor and approach used for estimation.**

We think reviewer mistakenly considered NCT (National Capital Territory) as NCR (National Capital Region). NCT is another name for Union Territory of Delhi and includes the seven districts of Delhi. On the other hand, NCR contains NCT and 19 other districts nearby to NCT. The generally quoted high emission figures represents the emis-

sions from NCR rather than NCT which is 20 times the area and contains thrice the population of NCT.

In the last decade Govt. of Delhi has taken serious steps to reduce the emissions in Delhi. It has closed almost all of the industries in Delhi, made CNG compulsory for Govt sponsored public transport etc. which have reduced the emission in NCT significantly.

**2  Reviewer #2 Comments**

AR2 raises some similar concerns to that of AR1. Further, the reviewer has highlighted the quality of language usage and writing style in the article, which we appreciate. However, common with AR1 they raise concerns about the general approach and methodology employed to generate the emissions at the higher resolution.

**I have only good words to say about this article. It is one of the easiest articles I have ever read and it flows very well for the reader. The science behind is novel and it would be a great tool for any modeller wants to include emission data from India to any transport, chemistry or climate model.**

Thank you for the kind words, we are glad to hear that it reads well for the reviewers.

**1) I am sceptic about the methodology used to create this spatial resolution, although the input was at a district level, hence, in course resolution. However, if we exclude inverse modelling approaches that would be able to create a fully resolved inventory, I admit this approach is the best we have.**

Unfortunately, we agree. The district-level data is the finest available presently, and hence why we feel it is important to include the uncertainty analysis and several approaches for calculation of the EFs and multiple assessments of activity data. Our approach has been similar to that employed by Qin and Xie (2011) for China, and we feel it offers a first-of-kind inventory for India that can be evaluated / employed in other

approaches.

Regarding the inverse methods, as discussed above, there have been a few approaches to employ such an approach to black carbon (e.g. Wang et al. (2016); Hakami et al. (2005). We feel however, that the level of uncertainty introduced in the meteorological modeling and furthermore chemical characterization and treatment of the black carbon aerosol in these approaches is as significant a source of uncertainty as the sub-district level distribution. Loss processes associated with black carbon aerosol behave non-linearly in the atmosphere, therefore it is a challenge to use an inverse approach. If one employs a lagrangian approach (e.g. Thompson and Stohl (2014)) they would violate assumptions of linear loss. Therefore, it is perhaps an improvement to use a full chemical transport model such as GEOS-CHEM (e.g. Fu et al. (2012)) or STEM (Hakami et al. (2005)). The challenge with such an approach is that these models utilize Eulerian discretization and the grid resolution create numerical effects that misrepresent the true fillamental transport processes associated with an aerosol like black carbon Sodemann et al. (2011).

**2) Another problem is the emission factors used in the study. I expect that the authors will justify more their choice of emission factor selection, since there are already-known and validated emission factors in the literature used for such purposes (showed also in Table 2). I am not sure if taking the mean emission factor would make sense since in some cases it varies a lot. In addition, I do not think that emission factors taken from Andrae and Merlet's paper (used in most models) can be given the same weight with other emission factor studies. Of course, since they do not validate their dataset, they do not need to evaluate the emission factors they used.**

A similar concern was also raised by AR1. Our choice of emission factors was aimed to appropriately account for the variation in fuel combustion technology across the vast geography of India. Fuel combustion technologies vary significantly especially in domestic fuel category (Which accounts for almost 50% of BC emissions in India).

This was one of the issues we wanted to address through the statistical approach in emission inventory development.

Most of our emission factors are derived from studies conducted in India , China or South Asia. In cases where we have derived factors from the global studies (Bond (2004); Streets (2003); Akagi et al. (2011); Ito and Penner (2005) etc.) we have almost exclusively used the emission factors explicitly given for India/South Asia/Developing country.

**3) I think that a missing point is evaluation of the database created from this work. I would highly recommend to the authors to try including this dataset to a global inventory and perform a few runs comparing the surface concentrations of BC from their model and the dataset with observations in Southeastern Asia. There are several measuring stations there and such a comparison would help all of us understand how valid this dataset is. At the moment, a poor discussion is performed for such an interesting topic. I believe that including the dataset to a model and performing the aforementioned analyses would expand the discussion a lot.**

Similar to (Qin and Xie, 2011) we do not include an evaluation. In fact, we find it interesting that the reviewer assumes we could simply conduct an analysis 'from their model'. We believe strongly such an analysis is not a casual affair and should be conducted in a robust fashion, particularly given the aforementioned challenges associated with black carbon. To do such a study correctly would be beyond the scope of this study. A tremendous amount of effort goes into the collection and identification of valid sources of information for the inclusion in an emission inventory and is not necessarily work that would be conducted by the same scientists best suited to conduct transport simulations studies.

That being said, we obviously intend and hope that such an analysis be conducted. It is exactly toward this ambition that we publish the inventory and we recognize that

**manuscript in its current form. I have seen several more subject-related manuscripts being rejected. However, I am glad he did not and the authors have the chance to provide a very useful database to the modelling community (after evaluation of their results, of course).**

We are also glad that the editor has not rejected the manuscript and provided us the opportunity to address the good comments provided by the reviewers. In particular, we have included greater detail regarding the distribution of the transport-sector emissions through the inclusion of a road network – a concern raised by both reviewers. We also feel the valuable comments and reviews have contributed to the discussion, and we have made some additions that address the reviewers concerns.

**References**

Akagi, S. K., Yokelson, R. J., Wiedinmyer, C., Alvarado, M. J., Reid, J. S., Karl, T., Crounse, J. D., and Wennberg, P. O.: Emission factors for open and domestic biomass burning for use in atmospheric models, Atmospheric Chemistry and Physics, 11, 4039–4072, doi:10.5194/acp-11-4039-2011, 2011.

Bond, T. C.: A technology-based global inventory of black and organic carbon emissions from combustion, Journal of Geophysical Research, 109, D14 203, doi:10.1029/2003JD003697, 2004.

Fu, T.-M., Cao, J. J., Zhang, X. Y., Lee, S. C., Zhang, Q., Han, Y. M., Qu, W. J., Han, Z., Zhang, R., Wang, Y. X., Chen, D., , and Henze, D. K.: Carbonaceous aerosols in China: top-down constraints on primary sources and estimation of secondary contribution, Atmospheric Chemistry and Physics, 12, 2725–2746, doi:10.5194/acp-12-2725-2012, 2012.

Hakami, A., Henze, D., Seinfeld, J. H., Chai, T., Tang, Y., Carmichael, G. R., , and Sandu, A.: Adjoint inverse modeling of black carbon during the Asian Pacific Regional Aerosol Characterization Experiment, Journal of Geophyiscal Research, 110, doi:10.1029/2004JD005671, 2005.

Ito, A. and Penner, J. E.: Historical emissions of carbonaceous aerosols from biomass and
fossil fuel burning for the period 1870-2000, Global Biogeochemical Cycles, 19, n/a–n/a, doi:10.1029/2004GB002374, 2005.

Kuenen, J. J. P., Visschedijk, A. J. H., Jozwicka, M., , and Denier van der Gon, H. A. C.: TNO-MACC II emission inventory; a multi-year (2003–2009) consistent high-resolution European emission inventory for air quality modelling, Atmospheric Chemistry and Physics, 14, 10 963–10 976, doi:10.5194/acp-14-10963-2014, 2014.

Qin, Y. and Xie, S.: Estimation of county-level black carbon emissions and its spatial distribution in China in 2000, Atmospheric Environment, 45, 6995–7004, doi:10.1016/j.atmosenv.2011.09.017, 2011.

Sodemann, H., Pommier, M., Arnold, S. R., Monks, S. A., Stebel, K., Burkhart, J. F., Hair, J. W., Diskin, G. S., Clerbaux, C., Coheur, P.-F., Hurtmans, D., Schlager, H., Blechschmidt, A.-M., Kristjánsson, J. E., , and Stohl, A.: Episodes of cross-polar transport in the Arctic troposphere during July 2008 as seen from models, satellite, and aircraft observations, Atmospheric Chemistry and Physics, 11, 3631–3651, doi:10.5194/acp-11-3631-2011, 2011.

Streets, D. G.: An inventory of gaseous and primary aerosol emissions in Asia in the year 2000, Journal of Geophysical Research, 108, 8809, doi:10.1029/2002JD003093, 2003.

Thompson, R. and Stohl, A.: FLEXINVERT: an atmospheric Bayesian inversion framework for determining surface fluxes of trace species using an optimized grid, Geoscientific Model Development, pp. 2223–2242, doi:10.5194/gmd-7-2223-2014, 2014.

Wang, P., Wang, H., Wang, Y. Q., Zhang, X. Y., Gong, S. L., Xue, M., Zhou, C. H., Liu, H. L., An, X. Q., Niu, T., , and Cheng, Y. L.: Inverse modeling of black carbon emissions over China using ensemble data assimilation, Atmospheric Chemistry and Physics, 16, 989–1002, doi:10.1029/2004JD005671, 2016.

Zhao, Y., Qiu, L. P., Xu, R. Y., Xie, F. J., Zhang, Q., Yu, Y. Y., Nielsen, C. P., Qin, H. X., Wang, H. K., Wu, X. C., Li, W. Q., , and Zhang, J.: Advantages of a city-scale emission inventory for urban air quality research and policy: the case of Nanjing, a typical industrial city in the Yangtze River Delta, China, Atmospheric Chemistry and Physics, 15, 12 623–12 644, doi:10.5194/acp-15-12623-2015, 2015.

---

## Referee Report (RR1)

Comments on ACP 2015-978 article:

Diesel generators (oil products consumed in buildings)

Fig. 8: comment on why the emissions are so high in northern Indian. Is it biomass heating?

Fuel balance method?

---

## Author Response (AR2)

**UiO : Department of Geosciences**
**University of Oslo**

Dr. Corinna Hoose
Editor
Atmospheric Chemistry & Physics
Copernicus Office

Date:     30 August, 2016
Your ref:   acp-2015-978

**Dear Dr. Corinna Hoose**
We are please to submit our response to the requested minor revisions of our manuscript,
"Monthly and Spatially Resolved Black Carbon Emission Inventory of India: Uncertainty
Analysis". Thank you for handling the editorial process and providing input on the required
changes to improve our manuscript.

We have addressed the concerns raised by the reviewers including conducting a Fuel Balance
assessment on the non-biomass burning emissions. Further, we evaluated and added the diesel
generators for small and distributed commercial operations. Lastly, regarding Figure 8, we
have added to the text to explain the reason for large emissions in the region.

We hope these modifications to the manuscript appropriately and adequately the concerns
raised by the reviewers, and look forward to publication of our manuscript.

A technical note; I have printed the revised version using the 'online' class, as it was the only
way I was able to get the tables to render properly. If you would prefer a version as 'acpd' (e.g.
without the two columns), I would be happy to make this as well.

Kind Regards,

John F. Burkhart
Associate Professor

encl: Abstract
encl: Final Manuscript
encl: Marked 'diff' version showing changes

[revised manuscript text omitted]